

# The December 2012 Mayo River debris flow triggered by Super Typhoon Bopha in Mindanao, Philippines: Lessons learned and questions raised

Kelvin S. Rodolfo[1], A. Mahar F. Lagmay[2,3], R. Narod Eco[3], Tatum Miko L. Herrero[3,4], Jerico E. Mendoza[2], Likha G. Minimo[3], and Joy T. Santiago[2]

[1]Professor Emeritus, Department of Earth and Environmental Sciences, University of Illinois at Chicago; Project NOAH consultant in 2013
[2](Philippine) Nationwide Operational Assessment of Hazards, Department of Science and Technology
[3]Volcano-Tectonics Laboratory, National Institute of Geological Sciences, University of the Philippines, Diliman
[4]Now at Magmatic and Hydrothermal Systems and Marine Geodynamics, GEOMAR - Helmholtz Centre for Ocean Research, Kiel, Germany

*Correspondence to*: Kelvin S. Rodolfo (krodolfo@uic.edu)

**Abstract.** Category 5 Super Typhoon Bopha, the world's worst storm of 2012, formed abnormally close to the Equator, and its landfall on Mindanao set the record proximity to the Equator for its category. Its torrential rains generated an enormous debris flow in the Mayo River watershed that swept away much of Andap village in New Bataan municipality, burying areas under rubble as thick as 9 meters and killing 566 people. Established in 1968, New Bataan had never experienced super typhoons and debris flows. This unfamiliarity compounded the death and damage. We describe Bopha's history, debris flows and the Mayo River disaster, then discuss how population growth contributed to the catastrophe, and the possibility that climate change may render other near-Equatorial areas vulnerable to hazards brought by similar typhoons. Finally, we recommend measures to minimize the loss of life and damage to property from similar, future events.

## 1 Introduction

Super Typhoon Bopha was the world's worst storm in 2012. In December of that year, its torrential rains on the southern Philippine island of Mindanao triggered an enormous debris flow in the Mayo River watershed that devastated *Barangay* (village) Andap in New Bataan, a municipality of Compostela Valley Province. Debris flows, although among the world's most destructive natural phenomena, remarkably are not generally well understood. Some textbooks classify debris flows as a type of landslide, but that term, when used as a synonym for "debris flow", makes most people mistakenly think of rock masses detaching off a cliff and accumulating near its base. Also mistakenly, debris flows are often called floods, mudslides or mudflows-- not only by the media, but by decision makers as well. In fact, the official descriptions of the disaster incorrectly termed and treated it a "flash flood", and relocation sites were initially evaluated in that context (MGB, 2012). In 2012 it was still not widely recognized that the conic-shaped alluvial fans with apices at the mouths of mountain gorges are built, over long periods by rarely-occurring debris flows, and thus are unsafe sites to occupy. Such lacks of understanding





may have tragic consequences for communities like Andap in mountainous terrain. To address this deficiency, we review debris flows and their deposits in general, and exemplify them with a detailed description of the Mayo River event.

Beyond the huge volume and rapidity of the flow itself, human factors contributed to this catastrophe. Such events are rare in Mindanao, and New Bataan was settled much too recently for its founders and inhabitants to be familiar with super typhoons and debris flows. It is worrisome that the rapidly growing Philippine population continues to expand into increasingly disaster-prone areas, and does so with insufficient hazard evaluation. Unregulated logging deforested the steep slopes, facilitating runoff, erosion and the land-sliding that fed the debris flow.

As part of Project NOAH (Nationwide Operational Assessment of Hazards), the Philippine government's disaster-assessment program, we studied the Mayo River debris flow until most of our resources and attention were urgently diverted to a major new Philippine disaster event. That was the world's worst storm of 2013, Super Typhoon Haiyan in December that generated the catastrophic storm surge which destroyed Tacloban City and damaged many other municipalities on the Visayan Islands, killing thousands of people.

Many questions about the Andap disaster still await our attention; in the interim this report describes for the larger community of disaster-mitigation specialists Super Typhoon Bopha, the Andap catastrophe and its detailed geologic bases. We review the historical role that population growth and insufficiently-guided settlement continue to play in generating "natural" disasters in the Philippines.

We then address the possibility that climate change will bring similar large storms and debris flows more frequently to Mindanao and to other sub-Equatorial areas similarly unused to them. We present the sparse record of tropical cyclones that made landfall on Mindanao since 1945, associated records of the Pacific El Nino-Southern Oscillation (ENSO), and all western North Pacific tropical cyclones from 1945 to 2014. Our review of the literature pertinent to the question is an invitation for commentary and advice from climatologists and meteorologists to guide our thinking as we proceed.

We describe our new program, an outgrowth of the Andap disaster, that has identified over 1,200 alluvial fan areas in the Philippines that are susceptible to debris flows and communities at risk from them. The program already has had significant successes. Finally, we discuss what else might be done to protect Mindanao and other vulnerable sub-Equatorial populations from climate-related hazards.

## 2 Super Typhoon Bopha

On 23 November 2012, a large area of convection began forming at 0.6°N latitude, 158°E longitude (NASA, 2012). Two days later, while still unusually close to the Equator at 03.6°N, 157°E it was categorized as a tropical depression (Fig. 1A).



On 26 November, while at 04.4°N, 155.8°E it was upgraded to Tropical Storm Bopha. It was too close to the Equator for the weak Coriolis effect there to develop its rotation quickly, but on 30 November, while still at 3.8°N 145.2°E, it was upgraded to a typhoon. Bopha intensified into a Category 4 Super Typhoon on 1 December while at 5.8°N, 138.8°E. On 2 December, it attained Category 5 wind speeds of 259 kilometers per hour while at 7.4°N, 128.9°E, the record proximity to the Equator for

that category. As it passed south of Palau Island on 3 December, Bopha weakened into a category 3 typhoon, and then re-intensified back to Category 5. It entered the Philippine area of responsibility at 8 a.m. local time on 2 December and was given the local name of Pablo.

On 4 December at 0445H, Bopha arrived at the eastern Mindanao coast at about 7.7°N (Fig. 1B), the landfall closest to the

Equator for all Category 5 tropical cyclones of record. Its average wind speeds and gusts were 185 and 210 kilometers per hour, respectively. Once onshore, Bopha weakened rapidly as it expended much of its energy by generating great havoc. Many fisher folk were lost at sea and many coastal dwellers were drowned. The National Disaster Risk Reduction and Management Council of the Philippine government (NDRRMC, 2012) also attributed numerous deaths and severe injuries to flying trees and debris, but by far the greatest cause of death and destruction was wreaked by the debris flow described in

this report that Bopha's intense rains generated (Fig. 1C). After passing through Mindanao, Bopha crossed the Sulu Sea and Palawan Island, entered the West Philippine Sea, and then reversed course towards northern Luzon, but dissipated before making landfall there.

On 12 February 2013, the United Nations Office for the Coordination of Humanitarian Affairs reported that while in the

Philippines, Bopha killed 1,146 people with 834 still missing, and displaced 925,412 others. It totally or partially damaged 233,163 houses and caused 1.04 billion U.S. dollars of damage; the most costly typhoon in the nation's history up to that time, only to be superseded by7 Super Typhoon Haiyan the following year.

### 3 Debris Flows

Among the world's most destructive natural phenomena, debris flows are fast-moving slurries of water and rock fragments, soil, and mud (Takahashi, 1981; Hutter et al. 1994; Iverson 1997; Iverson et al., 1997)). They can be triggered by sudden downpours such as commonly delivered by tropical cyclones, by reservoir collapses (Lagmay et al., 2007), or by landslides dislodged by earthquakes into streams. Many debris flows (Table 1) are associated with volcanoes (Vallance, 2000; Rodolfo, 2000; Lagmay et al., 2007). The size of a debris flow does not determine how much damage it will wreak and how many

people it will kill. In a poorly populated area such as Mount St. Helens, or where people are familiar with the hazard, such as lahars at Pinatubo Volcano, casualties can be light or even non-existent.

When rainfall on slopes exceeds critical thresholds of intensity, duration and accumulation it dislodges soil, sediment and rock masses into landslides that may coalesce to form debris flows -- slurries of sediment and water with the consistency of





freshly mixed concrete. Water contents rarely exceeds 25% by weight and may be only 10% -- just enough to provide mobility. Gravel and boulders constitute more than half of the solids, and sand typically makes up about 40%. Silt and clay normally constitute less than 10%, and remain suspended in the water (Pierson and Scott, 1985; Smith and Lowe, 1991).

While flowing in a channel, a striking debris-flow behavior is how easily it transports large boulders, owing only in part to the buoyancy provided by the density of the slurry. Boulders repeatedly bounce up from the channel floor or away from its sides into the central near-surface "plug" of the flow where friction with the channel is minimal and flow velocity is greatest. Thus, in a mountain gorge they tend to migrate to the front of the flow, where they add to a high, moving dam consisting largely of boulders, logs and tree debris.

Behind it, the moving frontal dam ponds the main flow body, which is richer in sand, silt and clay, and progressively becomes more dilute toward the rear, undergoing transitions into hyperconcentrated flows -- so-called because they carry much more sediment than normal streams do. Sand, silt and clay commonly comprise up to 75% by weight of hyperconcentrated flows, which look little different from normal, turbid flood waters, but flow twice as fast or more,

typically 2 to 3 meters per second (Pierson and Scott, 1985). Having no strength, they can transport gravel only as bed load. Hyperconcentrated flows in turn are succeeded by normal, turbid stream flow. Confusingly, "debris flow" sometimes refers to only a true debris-flow phase, and sometimes to an entire hydrologic event including its hyperconcentrated and normal stream-flow phases, which we do here in reference to the Mayo River debris flow.

Emerging from a mountain gorge, a debris flow spreads out, and increased basal friction slows it down. It drops some of its sediment load, adding to a conical alluvial fan, expressed on topographic maps by contour lines that are convex toward the downstream direction (Fig. 2). Even after it spreads out, it continues to transport large boulders by combined flotation, pushing, dragging and rolling. The flow may extend beyond the fan for many kilometers, especially its hyperconcentrated and normal-flood phases. Debris flows vary in volume by many orders of magnitude, from a thousand to a hundred thousand

cubic meters for the most frequent ones to more than a hundred million cubic meters (Table 1; Jakob, 2005). Importantly, debris-flow sizes correlate positively with velocities, which range from 2 to 100 kilometers per hour (Pierson, 1998; Rickenmann, 1999).

An important distinguishing characteristic of debris-flow deposits is "reverse grading": Boulders tend to be smaller at the

base and increase in size upwards. Large boulders commonly jut out at the top of a deposit (Fig. 3A). In addition to the buoyancy they experience from the dense slurry, the best mechanism advanced to explain reverse grading is *kinetic sieving* (Gallino and Pierson, 1985; Savage and Lun, 1988; Hutter et al., 1994; Vallance, 2000). While the flowing slurry is undergoing shear at its base, void spaces of different sizes continuously open and close, and particles of equivalent sizes migrate into them while they last. Smaller voids form more frequently and are filled by smaller solid particles, so larger



boulders migrate up to the flow surface. Debris-flow deposits are also characteristically "matrix-supported" (Fig. 3B); the larger rock fragments are separated by a mixture of the finer sediment that constituted the bulk of the flowing slurry that carried them. Pierson (2005) has published a useful guide for distinguishing the effects of debris flows from those of floods.

## 4 Methods

Prior to our field work, we mapped out the extent of the debris flow deposits using high-resolution optical satellite imagery acquired through Sentinel Asia, the collaborative initiative between space agencies and disaster management agencies that applies remote sensing and Web-GIS technologies to support Asia-Pacific disaster management. In the images, large boulders and other coarse debris easily discerned in the main debris flow body facilitated its delineation from the hyperconcentrated-flow deposits. The only available maps were 1:50,000 scale maps dating from the 1950's, and so we commissioned a LiDAR (Light Detection and Ranging) survey to generate detailed topographic maps of the affected areas for our fieldwork.

In the field, we analyzed the new deposits, ascertained that they were indeed left by a debris flow, and found evidence that enabled us to determine its velocity when it hit Andap. We also found and described old deposits which confirm that debris flows had happened long before New Bataan was established. The Bopha event was described for us in detail by residents and eyewitnesses we interviewed. We asked those who have lived in New Bataan since the 1960's whether similar events had happened before; they had not. Data gathered from these surveys and interviews were used to analyze and reconstruct the event.

## 5 Geomorphologic setting and history of New Bataan and the Mayo debris flow

Upstream of New Bataan and Andap, the Mayo River drains a mountainous watershed of 36.5 square kilometers with a total relief of about 2,320 meters and with slopes commonly steeper than 35° (Fig. 2). The Mayo River passes southward through a narrow gorge to join the Kalyawan River, which flows in the Compostela Valley that it shares with several other tributaries of the Agusan River.

Eight kilometers downstream of the Mayo junction, a site near the eastern edge of the Compostela Valley was informally known as Cabinuangan after its many enormous, valuable Binauang (*Octomeles sumatrana*) trees. This old-growth forest drew the attention of the logging industry in the early 1950s (Ea et al., 2013). As the loggers rapidly expanded their road networks, farmers from Luzon and the Visayan Islands followed closely behind, planting the cleared land mainly to coconuts, but also rice, corn, bananas, coffee, cacao, abaca and bamboo.

In 1966 the government subdivided the public lands of Compostela Valley into municipal areas, including one of 55,315 hectares that was further subdivided into farm lots and a 154-hectare central town site in Cabinuangan. When this new



municipality comprising 16 *barangays* was formally established by act of Congress on 18 June 1968, it was named New Bataan in honor of Luz Banzon-Magsaysay, the widow of President Magsaysay and a native of the Luzon province of Bataan, who had lent her influence to the town proponents. The central town retained "Cabinuangan" for its *barangay* name. In 1970, two years after its founding, the population of New Bataan was 19,978 (National Census and Statistics Office,
1970); by 1 May 2010 it had increased 238% to 47,470, including 10,390 in Cabinuangan and 7,550 in Andap (National Statistics Office, 2010).

Cabinuangan was laid out thoughtfully, with streets radiating out from a circular central core for government and social functions (Fig. 4A), but the founders of New Bataan were not informed about the natural hazards it faced. No one, including
anyone in government, realized that the Kalyawan River portion of Compostela Valley had served as an avenue for ancient debris flows. Indeed, debris flows were not widely understood at that time, and even the government-issued hazard map of New Bataan available in 2012 (MGB, 2009) was only concerned with landslides and floods. This lack of geomorphologic knowledge would prove fatal during super typhoon Bopha (Fig. 4B).

*Barangay* Andap was established at the head of the valley 3 kilometers upstream of Cabinuangan, on high ground that was not recognized as an alluvial fan but is clearly expressed as such in Figure 2 by contour lines that are convex downstream where they cross the valley. Characteristically reverse-graded, matrix-supported debris-flow deposits of unknown but ancient age built up the fan (Fig. 3).

**6 The Mayo River debris flow of 2012**
Rain-gauge data from Maragusan municipality 17 kilometers south of Andap are proxies for the rainfall that triggered the debris flow (Fig. 5). From midnight on 4 December until the flow occurred at 6:30 that morning, the Mayo River watershed above the alluvial fan received 120 millimeters of rain, falling as intensely as 43 millimeters per hour, and accumulated 4.4 million cubic meters. These values greatly exceeded the global initiation thresholds for debris flows, including those at the
Philippine volcanoes Mayon and Pinatubo (Rodolfo and Arguden, 1991; Westen and Daag, 2005), and Taiwan (Guzzetti et al., 2008; Huang, 2013).

After the debris flow began, it was sustained until 7 a.m. by another 24 millimeters of torrential rainfall that peaked at 52 millimeters per hour at 6:45 a.m. This delivered an additional 900 thousand cubic meters of runoff. Substantial discharge
from the 17.7 kilometer-square Mamada River watershed joined the debris flows about 450 meters downstream of the Mayo Bridge; this, along with discharge from other Kalyawan River tributaries, diluted the western portions of the debris flow into hyperconcentrated flows that reached 2 kilometers beyond Cabinuangan.



Other factors facilitated the debris flows. The rocks are extensively fractured because the watershed lies in the broad, left-lateral Philippine Fault zone of which the Mati Fault in Figure 2 is a major splay. Its steep slopes have been largely deforested by mining and logging, which facilitated numerous landslides, both shallow and involving bedrock, that were triggered by Bopha's heavy rains. Powerful typhoon winds uprooted trees on the upper watershed, enhancing infiltration-triggered soil slips and erosion by runoff, providing additional bulk that included clay, which increases debris-flow cohesion, mobility, and runout distance (Costa, 1984). Abundant, ancient, easily-remobilized debris-flow deposits underlay the path that the flows took (Fig. 3B).

At about 6:30 a.m., Andap resident Eva Penserga watched in horror as the 16-meter high front of a full-fledged debris flow emerged from the Mayo River gorge and obliterated a 100-meter long concrete bridge 1.5 kilometers upstream of Andap, carrying away a truck bearing 30 construction workers. Shortly thereafter, people in Andap witnessed the arrival of the debris flow, which lasted only about five to ten minutes. Unfortunately, alerts radioed the night before had directed about 200 people from outside Andap proper to seek shelter from floods at the community center where they joined many local inhabitants; 566 people were swept away, equivalent to 7.5% of the village population counted by the 2010 census.

Amateur video footage and the 5-kilometer length of the debris-flow deposit indicate a flow velocity of 60 kilometers per hour. No structures survived the main flow, but battered trees standing in the debris field 30 m from its eastern edge and 70 m upstream from the obliterated community center document slower flows there. The heights that the flows rose up against the trees yield their velocity (Arguden and Rodolfo, 1990): Assuming that all of the kinetic energy of the flow was converted to potential energy as it rose up against these obstacles, the 1.7 meter run-up height $h$ indicates a velocity $v$ of 5.8 meters per second or 21 kilometers per hour, from $v = (2gh)^{1/2}$. This value is only minimal, because the formula considers neither channel roughness nor internal friction.

From satellite imagery and our post-Bopha LiDAR mapping and field measurements, the volume of the Andap debris-flow deposit is 25 to 30 million cubic meters, ranking it among the largest ever experienced worldwide (Table 1). The deposit is 0.2-1 kilometer wide and 0.25 to 9 meters thick. Debris with boulders up to 16 meters in diameter (Fig. 3C) covers 500 hectares and buried Andap as much as 9 meters. Downstream, clast sizes decrease and the deposits thin, grading into sandy, laminated hyperconcentrated-flow deposits less than 0.5 meter thick. These finer-grained deposits cover 2000 hectares and extend 8 kilometers beyond Cabinuangan. Where these finer-grained sediments dominate, associated tree debris clogs streams and creeks. In Cabinuangan, dozens of corpses were recovered from a tangle of fallen trees and logs (Fig. 3D).

## 7 The role of Philippine population growth

The global increase in death and damage from natural calamities may be due in part to the effects of anthropogenic climate change, but another, more likely reason is the growth of populations in high-risk areas (Huppert and Sparks, 2006),



especially in places that experience tropical-cyclone landfalls (Weinkle et al., 2012) . Nowhere is this better exemplified than in Mindanao, and by the Andap disaster.

The founding of the newer Mindanao settlements including New Bataan was largely driven by the pressure of rapid
population growth, well described by Dolan (1993). In 1950 the Philippine land-population ratio was about one cultivated hectare per agricultural worker; by the early 1980s the ratio had been cut in half.   The 1980 census documented that six of the twelve Philippine provinces experiencing the fastest growth were in western, northern and southern Mindanao.  In the early 1980s, this population growth slowed down because of increasing friction and armed conflict with Muslim and other indigenous tribes.  Furthermore, the leftist New Peoples' Army had also expanded into Mindanao and began sporadic
fighting with the Philippine Army.  The general lawlessness was enhanced by banditry, including by entire government army units of deserters, which Filipinos referred to as "lost commands".

When New Bataan was settled in 1968, the annual Philippine population growth rate was 2.98 percent and Filipinos numbered 36,424,000. By 2014 the population had almost tripled, to 107,668,000 (United States Census Bureau
International Programs, 2014). A Reproductive Health Care congressional bill was filed in 2003, its main purpose being to provide contraception to the poor. After strenuous opposition from the clergy in this predominantly Roman Catholic country, the bill was finally passed in December 2012, coincidentally the month that Bopha arrived. The annual growth rate has dropped to 1.83 percent, but that means that the country will still need to provide for another two million people in 2016, and similar numbers every year for some time to come. Among these needs, housing will be extremely difficult to find because
hardly any hazard-free areas remain.

About two-thirds of the Philippine population live, farm, and catch and grow fish in coastal areas, which are developed and crowded to capacity.  Metro Manila, the largest metropolitan area, is extracting so much groundwater that its coastal plains are subsiding several centimeters to more than a decimeter annually, losing area to the sea and becoming ever less able to
accommodate more people because of worsening floods and tidal incursions (Rodolfo and Siringan, 2006).  The very real possibility that other rapidly-growing Philippine coastal cities are experiencing the same problems still awaits investigation.

The problems of Metro Manila are exacerbated by the continued influx of more than 900,000 informal settlers every year (Vicente et al., 2005). Large corporations are taking advantage of the urgent need for living space by seeking permission
from the government to reclaim 38, 272 hectares of coastal offshore areas, 26,234 along Manila Bay alone.   This, even though Manila Bay is subsiding rapidly, is subject to storm surges, and is overdue for a major earthquake that would disproportionately damage reclaimed land by enhanced ground shaking and liquefaction (Rodolfo, 2014). Inexorably, other people are seeking living space inland, where natural hazards abound.



In February 2013 the office of the Philippine President organized Task Force Pablo, a multi-agency group of geologists and engineers of the Mines and Geosciences Bureau (MGB) and Project NOAH, to conduct field analyses of the Andap disaster and search for safe relocation sites for the people of New Bataan and other municipalities of the Province of Compostela Valley. Task Force Pablo identified 31 resettlement sites, using LiDAR-derived digital terrain models and rainfall intensity-duration frequency data from the national weather service. The phenomenal event at Barangay Andap required special attention from us to identify relocation sites safe from future debris flows in New Bataan. The task is a daunting one; the Kalyawan floodplain is susceptible to floods and debris flows, the valley margins and adjacent high grounds, to landslides.

## 8 Is Bopha a harbinger of the future?

### 8.1 The historical record of tropical cyclone landfalls in Mindanao

The ancient debris-flow deposits in New Bataan testify that such flows occurred in Compostela Valley at least once before Super Typhoon Bopha. Dating those deposits is a prime topic for future research. At present, all we can say is that the event occurred long before New Bataan was settled in 1968; the sizes of some trees rooted in the old deposits suggest decades or even a century or more earlier.

The most urgent question raised by these old deposits and by the Andap disaster is whether their debris flows simply represent the latest, very rare and essentially random events in Mindanao, or whether it and other places at low latitudes can expect to experience them more frequently as the climate changes. Most climatologists (Webster et al., 2005; Emanuel, 2005; Bengtsson et al, 2007; Elsner et al., 2008; Emanuel et al., 2008; Knutson et al.,2010) equate climate change with fewer but more intense tropical cyclones due to rising sea-surface temperatures and atmospheric water-vapor contents. But this does not necessarily mean that typhoons will make Mindanao landfall more frequently in the future, even though their history since 1945 might suggest as much (Fig. 6A).

Tropical cyclones rarely and sporadically make landfall on Mindanao because the island lies in the ephemeral southern fringe of the northwest Pacific typhoon track. Furthermore, most Mindanao typhoons do not occur during the main season of July through October, and most are tropical depressions, hence do not enter into most modeling attempts to predict future typhoon behavior.

Miguel Selga (1935), a Spanish Jesuit, compiled a chronology of typhoons that made landfall in the Philippines from 1566 to 1900. García-Herrera et al. (2007) have critically analyzed this chronology, counting 520 typhoons and 102 weaker tropical storms. They provided location charts from Selga's data that show only 20 affecting Mindanao. Given the many uncertainties that Selga faced, this compilation is of limited value, particularly for Mindanao. Aside from their enclave of Zamboanga, the Spaniards had little success in occupying and governing Mindanao by subjugating the indigenous Muslims.




Selga's data of more than three centuries does qualitatively confirm that Mindanao has been much less affected by tropical cyclones than the rest of the Philippines.

In 1945 the U.S. Navy Joint Typhoon Warning System began to archive northwest Pacific tropical cyclones, recording only 34 Mindanao landfalls by the end of 2012 (Unisys Weather, 2012). A tropical depression arrived in January 2013. On 13 January 2014 Tropical Depression Lingling (local name Agaton) made landfall and killed 70 people on Mindanao (NDRRMC, 2014). On 29 December 2014, Tropical Storm Jangmi (NASA, 2014) killed ten people in Mindanao (NDRMMC, 2015a). Finally Tropical Depression Onyok arrived on 18 December 2015 (NDRRMC, 2015b). These 38 landfalls are incontrovertible, and our search for what the future holds begins with them.

During 40 of those 69 years, not even a single tropical depression visited the island; one quiescent period lasted eight years, from 1956 to 1963 inclusive. Most of the tropical cyclones that affected Mindanao were of the weaker varieties: 21 tropical depressions, 11 tropical storms, and Category 1 typhoons Violet in 1955 and Lola in 1975.

Before Bopha, Mindanao was largely spared stronger typhoons except for Category 5 Louise in 1964, Category 3 Kate in 1970, and Category 4 Ike in 1984. Louise and Ike both barely grazed the northernmost tip of the island, and Kate passed some 45 kilometers south of New Bataan, where it is remembered as not being very windy, only for its heavy rains and flooding. Only four tropical cyclones of all categories arrived during the northwest Pacific peak typhoon season of July through October, although these included Kate in October 1970 and Ike in September 1984. Twelve came in pre-seasonal March through June, and nineteen arrived during the post-season months of November through January.

From 1945 to 1989, the frequency of Mindanao landfalls was only one every 2.5 years. Then that rate abruptly doubled, to one landfall every 1.32 years in the period from 1990 to 2015. Another fact causing concern is that Mindanao for the first time has recently suffered lethal cyclones in two consecutive years. The year before Bopha, on 16-17 December 2011 the city of Cagayan de Oro on the Mindanao coast 180 kilometers north of New Bataan received 180 millimeters of rain from Tropical Storm Washi. Most fell during only six hours, causing floods that killed 1,268 people (Ramos, 2011; Manila Observatory, 2012). A tropical depression made landfall on Mindanao two months before Washi, so 2011 was only the fifth year since 1945 for Mindanao to experience two tropical cyclones. Only three years later in 2014, Mindanao again experienced two lethal tropical cyclones: Tropical Depression Lingling and Tropical Storm Jangmi.

The increase in Mindanao storminess since 1990 is striking and alarming. It cannot be ascribed simply to the climate change induced by anthropogenic global warming, however. It must be evaluated in the context of significant multi-annual and multi-decadal fluctuations in sea-surface temperatures in the western North Pacific of the Pacific Decadal Oscillation (PDO) and the El Niño – Southern Oscillation or ENSO, as well as multi-decadal fluctuations in tropical-cyclone frequencies.



## 8.2 Complicating oscillations: PDO and ENSO

The Pacific Decadal Oscillation or PDO is so-called because its periods last for two or three decades (Mantua and Hare, 2002; JISAO, 2013). Cool PDO conditions lasted from 1890 to 1924 and from 1947 to 1976; warm PDO regimes occurred during 1925 to 1946 and from 1977 to about 1990, after which no clear dominance is apparent. PDO changes are most pronounced north of 20ºN where the phenomenon was discovered and where it seriously affects weather and oceanic life, including commercial salmon abundances. PDO conditions are much more subtly expressed in the tropics; not surprisingly, they do not seem to correlate with Mindanao landfalls, or even with the frequency of all northwest Pacific typhoons.

ENSO is the complex result of ocean-atmosphere interactions that are best expressed by fluctuating sea-surface temperatures in the central and eastern equatorial Pacific, from warmer during El Niño to cooler during La Niña periods (Trenberth, 1997; Wolter and Timlin, 2011). Atmospheric pressures at the ocean surface during an El Niño are high in the western Pacific and low in the eastern Pacific, and the situation is reversed during a La Niña. Typical episodes of both occur every three to five years, but El Niños tend to last nine months to a year, La Niñas, one to three years (NOAA Climate Prediction Center, 2014).

During El Niño episodes, tropical cyclones tend to form farther east, are more widely dispersed, and curve northward, making fewer Philippine landfalls. Under La Niña conditions they tend to form farther west, stay below 23ºN, and travel westward, thus visit the Philippines more frequently, especially during the later typhoon months of September through November (Wang and Chan, 2002; Wu et al. 2004; Emanuel, 2005; Emmanuel et al., 2008, Zhang et al., 2012). Except for their tendency to arrive later than November, all the typhoons before Bopha that made Mindanao landfalls since 1945 fit that pattern by occurring during La Niñas (Fig.s 6A and 6B). Bopha came either during a weak La Niña (NOAA Climate Prediction Center (2014), or weak El Niño (NOAA Earth Science Research Laboratory, 2014). The weaker storms and depressions visiting Mindanao show no marked preference between El Niño and La Niña episodes; 15 vs. 22, respectively.

Cai et al. (2015) recently analysed 21 global-climate models of Phase 5 of the Coupled Model Intercomparison Project commissioned by the Intergovernmental Panel on Climate Change. They have arrived at the disquieting conclusions that global warming will double the frequencies of future extreme La Niñas, from the historical average of one every 23 years to 13 years. They ascribed the change to three effects of global warming. The western North Pacific region of archipelagos and insular seas that includes the Philippines will warm faster than the central Pacific; vertical temperature gradients of the upper tropical ocean will be enhanced; and extreme La Niñas usually follow extreme El Niños, which will also occur more frequently (Cai et al., 2014). Given the tendency of typhoons to make landfalls on the Philippines more frequently during La Niñas, the country including Mindanao should expect greater storminess in future.

## 8.3 Fluctuations in annual tropical-cyclone frequencies



Annual frequencies in cyclone occurrences also fluctuate significantly over lengthy periods. Figure 6C presents two compilations of the yearly numbers of tropical cyclones in the northwest Pacific. Matsuura et al. (2003), analyzing data archived by the Tokyo Typhoon Center, reported cycles of fluctuating annual numbers lasting about two decades with about ten more tropical cyclones per year during peaks than during troughs. Numbers were low from 1951 to 1961, high from 1961
to 1972, low again from 1973 to 1985, and high from 1986 to 1994. A low-frequency period started in 1994. These researchers ascribed the cyclicity to long-term sea-surface temperature variations in the tropical central Pacific, coupled with westerly wind anomalies that come with the monsoon trough which develops during the main typhoon season of July to October.

Liu and Chan (2013) modified and updated this study, but instead used data archived by the U. S. Navy's Joint Typhoon Warning Center (Unisys Weather, 2012), starting only in 1960 when satellites afforded improved positioning. Their high-frequency periods are 1960 to 1975 and 1989 to 1997, with low-frequency intervals from 1974 to 1987 and 1998 to 2011.

Neither Matsuura et al. (2003) nor Liu and Chan (2013) counted tropical depressions among their data because these are
harder to define and are more ephemeral. This omission, however, eliminates 21 of the 38 Mindanao landfalls. Landfalls are easy to locate, however, and they can be lethal on Mindanao, as most recently demonstrated by Lingling in January 2014. Accordingly, in Figure 6C we augment the data of Matsuura et al. (2003) and Liu and Chan (2013) with a histogram of all tropical cyclones, including tropical depressions, recorded by JTWC since 1945. This does increase some annual numbers, but does not change the general pattern and timing of the fluctuations compiled earlier.

The report by Liu and Chan (2013), entitled 'Inactive Period of Western North Pacific Tropical Cyclone Activity in 1998–2011', exemplifies the lack of attention to low-latitude areas like Mindanao. In addition to ignoring tropical depressions, they also limited their data set to tropical cyclones of the June to October main typhoon season. Only one of the twelve Mindanao landfalls arrived during those months from 1998 to 2011. For Mindanao, unlike for the northwest Pacific as a
whole, that period was not at all a slack period for tropical cyclones (Fig. 6A).

### 8.4 Other models of future typhoon behaviour

A recent review by ten prominent researchers studying the long-term response of tropical cyclones to climate change (Knutson et al., 2010) stated that considerable research on the topic has yielded conflicting results because of large
fluctuations in cyclone frequencies and intensities, as well as serious deficiencies in the availability and quality of historical records. Thus, it is uncertain whether the observed changes in tropical-cyclone activity exceed the variability due to natural causes. The authors do have some confidence in theory and models that project globally averaged frequencies of all tropical cyclones to decrease 6-34% by 2100, but for intensities to increase 2-11% owing to substantial increases in the most intense





cyclones. Most worrisome for debris-flow generation, the review predicts that precipitation within 100 kilometers of storm centers will increase about 20%.

Kossin et al. (2014) analyzed the lifetime maximum intensities of the typhoons that occurred during the August-October peak season from 1982–2012, for which satellites have gathered a wealth of precisely located physical detail. They made a convincing statistical case that the typhoon track has migrated pole-ward 53 kilometers per decade during that 31-year period. Again, this finding has little bearing on the history and future of Mindanao typhoons, which mostly occur off-season.

Bengtsson et al. (2006) have reported a recent strengthening and equator-ward shift of the May through October typhoon tracks in the eastern Pacific. This cannot be equated with the western Pacific, although Wu and Wang (2004) have predicted that the tracks there will shift toward the Equator during the period 2000-2029. A model by Brayshaw et al. (2006) finds that storm tracks shift toward the Equator in response to increased sea-surface temperature gradients in the tropics. This is an "aquaplanet" model, one of an Earth without continents, and is therefore of debatable value, although experiments

employing a 2K increase appear to agree (Graff and LaCasce, 2012). El Niño warming of the equatorial Pacific also seems to cause a similar shift (Chang et al., 2002; Lu et al., 2008). Another study of the potential effect of global warming on tropical cyclone activity (Li et al., 2010) used a global high-resolution atmospheric general circulation model to conclude that tropical cyclone locations will shift significantly from the western to the central Pacific. To arrive at that conclusion, however, they diagnosed the dynamic and thermodynamic conditions in the July–October main typhoon season. This, too,

does not address what might happen in low-latitude areas like Mindanao that experience tropical cyclones during the off-season.

In short, the record of increasingly frequent landfalls on Mindanao may or may not indicate that more frequent typhoon disasters will happen there in the future, although the results of Cai and coworkers (2014, 2015) strongly suggest as much.

Low-latitude areas, however, are given short shrift by most meteorological and climatologic analyses. Given the large populations that live near the Equation, more research of the possible impact of anthropogenic global warming on tropical cyclone behavior there is urgently needed.

### 8.5 The NOAH national catalog of alluvial fans and debris flow susceptible areas

A positive outgrowth of the Andap disaster is the compilation by NOAH of all alluvial fan areas in the Philippines (Aquino et al., 2014). Alluvial fans were delineated from high-resolution digital terrain models by analyzing geomorphic features, slopes, gradients, and stream networks. So far, more than 1,200 alluvial fans have been identified throughout the country, and communities under the threat of debris flows are being educated about them. The results can be accessed online for free in the NOAH portal at http://noah.dost.gov.ph.



In October 2015, Typhoon Koppu (Lando) generated devastating debris flows on alluvial fans in Nueva Ecija province (Eco et al., 2015). Fortunately, communities living on those alluvial fans had been warned and evacuated. No one was killed.

In December 2015, Typhoon Melor (Nona) struck Mindoro in the central Philippines, also triggering massive debris flows. Houses and buildings were buried or washed out in several communities on alluvial fans, but no one died because of timely warnings and evacuations (Llanes et al., 2016).

## 8.6 Other climate-related hazards in the Philippines and Mindanao

Future fluctuations between extreme El Niños and La Niñas pose other threats. Philippine rainfall is modulated by ENSO; El Niños bring droughts and La Niñas cause excessive rainfall (Lyon et al., 2006). During a protracted El Niño drought, rock debris accumulates on slopes that heavy rains of the succeeding La Niña wash down, causing landslides and debris flows. Additionally, excessive La Niña rainfall encourages strong forest growth that a succeeding protracted drought dries out and renders susceptible to fire.

Mindanao has nine active and twelve potentially active volcanoes (PHIVOLCS, 2008) that are popular tourist destinations, productive geothermal areas, and mining districts. Many are situated in watersheds with important agriculture and large populations. However, like Mt. Pinatubo on Luzon Island before its disastrous 1990 eruption, these volcanoes have not yet been fully studied or instrumentally monitored, and their populations are unused to eruptions. As Table 1 shows, some of the

world's largest debris flows are lahars generated on volcanoes by intense rainfall during an eruption or even decades afterwards. Whether or not typhoons will visit Mindanao more frequently in future, any large eruption there will inevitably be succeeded by a major storm sooner or later. Even without eruptions, Mindanao's larger, taller volcanoes pose serious threats, being structurally and mechanically weak (Herrero, 2014) and thus susceptible to landslides and debris flows during exceptionally strong rainstorms.

## 9 Summary and Conclusions

Bopha formed abnormally close to the Equator. It developed into a Category 5 Super Typhoon and made landfall at record proximities to the Equator for all tropical cyclones of that category anywhere in the world. In only seven hours, it delivered more than 120 millimeters of rain to the Mayo River watershed, generating a debris flow that deposited a dry volume of 30

million cubic meters, the world's seventh largest of record. The village of Andap was devastated and 566 of its inhabitants were killed.

Debris flows are among the most lethal of natural hazards. They are remarkably poorly recognized in the Philippines -- especially in Mindanao, which lies in the southern fringe of the western North Pacific typhoon track and thus has been





infrequently visited by typhoons and debris flows. This unfamiliarity exacerbated the loss of life from the Mayo River debris flow.

"Every health centre or school that collapses in an earthquake and every road or bridge that is washed away in a flood began
as development activities" (UNDP, 2004). The people and government authorities who established New Bataan and Andap in 1968 did not know that they were building on ancient debris-flow deposits, and were unaware of the hazardous process that produced them. The lack of awareness about debris flows persisted until Bopha approached, when many people were advised to seek refuge from flooding on high ground at Andap. Even after the disaster, the government personnel initially designated to explain the tragedy and select relocation sites treated it as a "flash flood", not as a debris flow (Mines and
Geoscience Bureau, 2012).

The rapid growth of the Philippine population provided the impetus for the establishment of New Bataan and Andap in the late 1960s. A Reproductive Health Care congressional bill filed in 2003 was finally passed in 2012; how successful it will be in curbing population growth remains to be seen. Meanwhile, the population continues to expand into more areas
susceptible to natural hazards. Drawing upon the Andap and numerous other recent disasters, the government must more rigorously assess the hazards posed to new settlement sites and infrastructure.

Western North Pacific tropical cyclone data have been archived accurately since 1945. The frequency of Mindanao landfalls has doubled since 1990, a possible indication that anthropogenic global warming is making such events more frequent.
Learning whether this true or not is obscured by irregular climatic rhythms on the ENSO time scale of a few years in the western North Pacific. Additionally, most tropical cyclones that affect Mindanao do not arrive in the main typhoon season of July through October, and most are only tropical depressions, which most climatologists and meteorologists do not include as data for their models. The typhoon regimens of Mindanao and other, more densely populated low-latitude areas need more attention.

Typhoons make Philippine landfalls most frequently during La Niña episodes during a July-October main season. In Mindanao, however, they arrive during the off season from November to June. Current models suggest that extreme El Niños and La Niñas will succeed each other more frequently, a prime example of how Earth systems, kept in balance by myriads of interacting phenomena, fluctuate strongly when disturbed. Thus, Mindanao and the Philippines as a whole should
prepare their populations for more frequent hazards associated with these events, including landslides, debris flows and forest fires.

Developing countries have difficulty funding mitigation measures, and the best and least costly recourse is to enable each family to develop its own emergency plans, with accurate, accessible, understandable and timely government input. Among





NOAH's mandated tasks are to evaluate the numerous natural hazards that confront every region of the Philippines, to educate every community about the hazards they face, and to advise them how to prepare to protect themselves when the threats materialize. Thus, as a consequence of our work on the Mayo debris flow, Project NOAH has examined detailed topographic maps for the entire Philippine archipelago and identified more than 1,200 alluvial fans and associated
communities that may be threatened by debris flows (cite). We have also simulated potential flow paths of debris flows on all the alluvial fans and identified communities threatened by them.

### Acknowledgements

This work was funded by the Philippine Department of Science and Technology (DOST) and the Volcano Tectonics
laboratory of the National Institute of Geological Sciences at the University of the Philippines (U.P.). LiDAR data covering the New Bataan area were provided by the U.P. Training Center for Applied Geodesy and Photogrammetry. DOST's Balik (Returning) Scientist Program funded KS Rodolfo's travel. We thank Eric Colmenares for helping coordinate our field work, Jen Alconis, Yowee Gonzales, Jasmine Sabado, and Yani Serrado for their help in the field, DOST's Advanced Science and Technology Institute and the Philippine Atmospheric, Geophysical and Astronomical Services Administration for rainfall
data, Congresswoman M. C. Zamora for logistical support, and Thomas Pierson for information about debris-flow mechanics.

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

**Figure 1: Typhoon Bopha (Pablo). A = Track and development of the super typhoon. New Bataan, Andap and the Maragusan rain gauge lie beneath the Category 3 icon following Mindanao landfall. B = Bopha at landfall (modified from NASA Earth Observatory, 2012). C = Tropical Rainfall Measurement Mission (TRMM) image from which NASA (2012) estimated that Bopha delivered over 240 millimeters of rainfall near the coast.**





**Figure 2: Physical setting of the Andap disaster. Grey area enclosed by dashes is the Mayo watershed. All steep slopes are contoured at 50-meter intervals. Below 700-meter elevations the contour interval is 20 meters to better define the gentler valley surfaces. New deposits of true debris flows south of the Mayo Bridge are shown in solid black; associated hyperconcentrated-flow**



deposits are shaded in grey. Note that the topographic contour lines from the Mayo Bridge to New Bataan are convex northward, defining the surface of an alluvial fan. The trace of the Mati Fault is only generalized; it has numerous associated fractures in a broad zone along its length.

Figure 3: Debris-flow deposits in the New Bataan area. A= Boulder in ancient reverse-graded debris-flow deposit. Well-established trees indicate an age of some decades prior to the settlement of the town. B= Old debris-flow deposits underlying New Bataan – Andap highway. Boulders are separated from each other by a matrix of finer-grained sediment. For scale, the concrete is 15 centimeters thick. The coarse sediment atop the highway are new debris-flow deposits. C= Boulder-rich deposits of debris





flows that destroyed much of the barangay. D= Tangle of fallen trees and branches left with numerous cadavers by hyperconcentrated flows in central New Bataan.

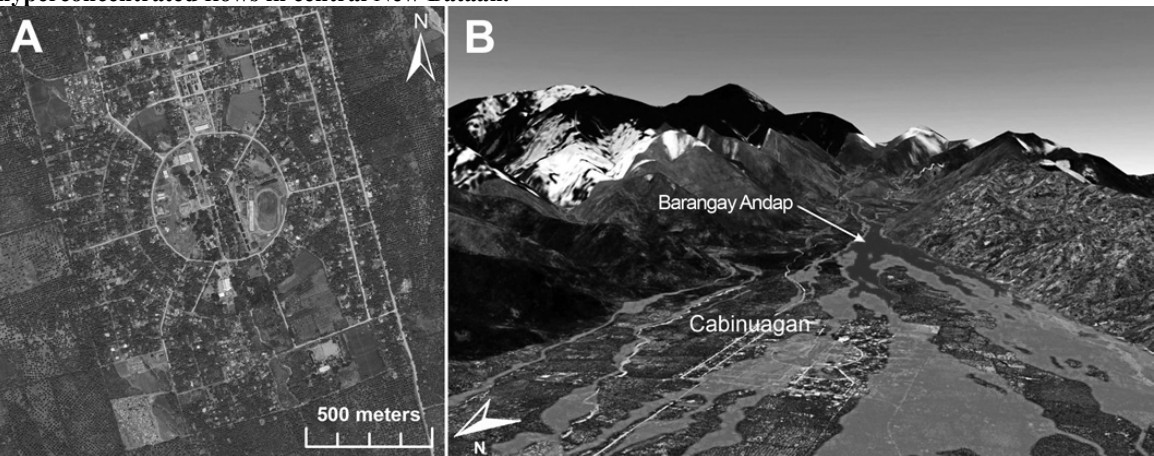

Figure 4: New Bataan. A = Google image of Cabinuangan (the central district of New Bataan) before the debris flow. B = Southward facing three-dimensional terrain diagram of New Bataan, showing Cabinuangan and the site of outlying Barangay Andap.

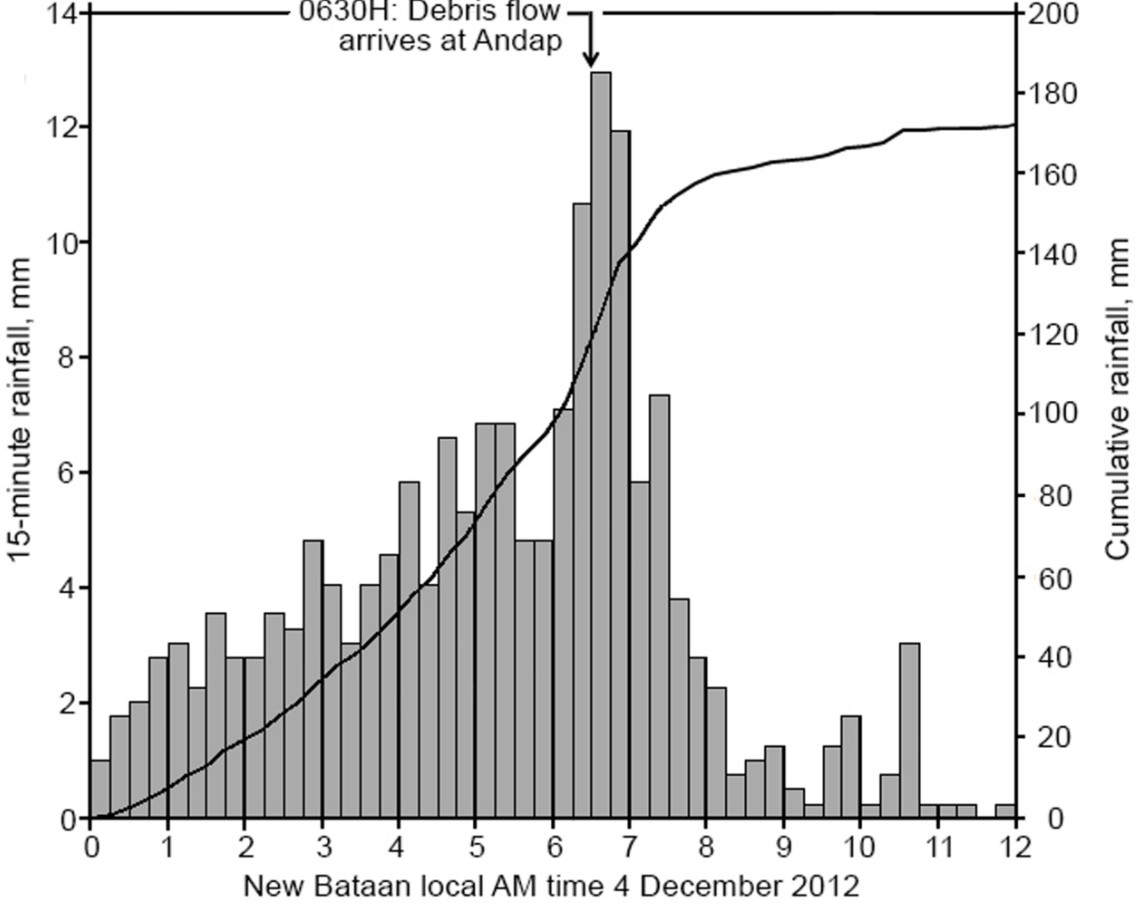



**Figure 5: The rainfall that triggered and sustained the debris flows. Histogram measures rain accumulated during successive 15-minute intervals; the heavy curve is accumulated rainfall.**



5  **Figure 6: Figure 6A from Unisys (2011), JTWC (2012, 2013), NASA (21 January 30 December 2015). TD = Tropical Depression; TS = Tropical Storm. Typhoons fully dated; TDs and TSs dated by month number only (January = 1 to December = 12). 6B modified from NOAA Earth Science Research Laboratory (2014). 6C: 1950-1959 data from Matsuura et al. (2003); shaded portions and frequency characterizations from Liu and Chan (2013); 1945-2015 histogram compiled from Unisys (2011), JTWC (2012, 2013), NASA (2014) and NDRRMC (2015b).**

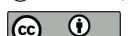



| Location | Date | Trigger | Volume in million cu. meters | Deaths | Reference |
|---|---|---|---|---|---|
| Rios Barrancas and Colorado, Argentina | 1914 | Failure of ancient landslide dam | 2,000 (estimate) | ? | Schuster et al., 2002 |
| Bucao River, Pinatubo Volcano, Philippines | 10 July 2002 | Caldera lake breach | << 160 | 0 | Lagmay et al., 2006 |
| Bucao River, Pinatubo Volcano, Philippines | 5-6 Oct 1993 | Typhoon Flo (Kadiang) rains | 110 | 0 | Remotigue, 1995 |
| Kolka Glacier, North Ossetia, Russia | 2002 | Large glacial detachment | 100 | 125 | Haeberli et al., 2004 |
| Nevados Huarascan, Peru | 1970 | Pyroclastic flows melted snow & ice | ~100 (flow volume) | 18,000 | Plafker & Erickson, 1978 |
| Nevado del Ruiz, Colombia | 13 Nov 1985 | Pyroclastic flows melted snow & ice | 40 | 23,000 | Schuster et al., 2002 |
| Mayo River, Mindanao, Philippines | 4 Dec 2012 | Typhoon Bopha (Pablo) rainfall | 25-30 | 566 | This report |
| Cordillera de la Costa, Vargas, Valenzuela | Dec 1999 | Heavy rain | 19 | 30,000 | Wieczorek. 2002 |
| Mayon Volcano, Philippines | 30 Nov 2006 | Typhoon Durian (Reming) rains | 19 | 1,226 | Paguican et al., 2009 |
| Pine Creek-Muddy River, Mount St. Helens, Washington, USA | 18 May 1980 | Pyroclastic surge melted snow and ice | 14 | 0 | Pierson, 1985 |

Table 1: The world's ten largest debris flows of record, ranked by volume.