# Peer review of "The December 2012 Mayo River debris flow triggered by Super Typhoon Bopha in Mindanao, Philippines: Lessons learned and questions raised"

_Natural Hazards and Earth System Sciences, 2016_

## Referee Comment (RC1) · Anonymous Referee #1 · 12 May 2016

GENERAL COMMENT

I've reviewed the paper "The December 2012 Mayo River debris flow triggered by Super Typhoon Bopha in Mindanao, Philippines: Lessons learned and questions raised", by Kelvin S. Rodolfo and co-authors. The paper presents several information about the Typhoon Bopha that struck southeastern Asia on December 2012 and on all the related ground phenomena, and damage.

The description of the typhoon is accurate, also from a meteorological point of view. Detailed information about typhoon and related phenomena are provided, together with

some personal communication from eye-witnesses. The paper is clear, understandable, well-written, and with a distinct structure, although it is too long in some parts.

Nevertheless, the main drawback of this paper, in my opinion, is that it is not a scientific research paper. According to the definition reported in the NHESS journal website, a research paper should report "substantial and original scientific results". This is not the case of this work, which does not propose any original scientific result, any new method, but only a complete description of the phenomena and several considerations about the past a future evolution.

The paper could be classified as a (very interesting and well-compiled) technical report. In fact, nothing about new methods or insights to deal with typhoons and related triggered debris-flows is presented or proposed in the paper.

I suggest to reduce a lot the text, in order to fit the directions of NHESS journal for remaining into 4 journal pages, and thus, to re-submit the paper in the form of "Brief communication". The topic fits perfectly the (c) point of the "Brief communications" descriptions, given that it "disseminate information and data on topical events of significant scientific and/or social interest within the scope of the journal".

SPECIFIC COMMENTS

Regarding the classification of the debris flows, I agree with the Authors when they state that debris flows could not be classified as floods. Nevertheless, at page 1, rows 25-27, Authors state that: "Some textbooks classify debris flows as a type of landslide, but that term, when used as a synonym for "debris flow", makes most people mistakenly think of rock masses detaching off a cliff and accumulating near its base." First, this sentence seems quite unclear to me. Second, Authors should acknowledge that, in addition to textbooks, the most known classifications of landslide types (i.e., Varnes, 1978; Cruden and Varnes, 1996), recently updated by Hungr et al. (2013), do consider debris flows as a type of landslide.

Concerning the description of the debris flow and of the other triggered phenomena, Authors state that they "mapped out the extent of the debris flow deposits using high-resolution optical satellite imagery acquired through Sentinel Asia" and "in the field, analyzed the new deposits, ascertained that they were indeed left by a debris flow, and found evidence...". Nevertheless, what I can see from the figures and the results is only a contour of the deposit of the triggered debris flow at a very coarse scale, in Figure 2, and some photos of the ground phenomena.

Lines from 5 to 25, in page 4, could be reduced or removed. Section 5 is very long and not so useful, expect for some parts, to the discussion proposed in the paper. I suggest revising this section and reducing it. Some historical notes (from page 6, line 27 to page 6, line 13) could be removed from the text.

The description of the event, in section 6, is very detailed and interesting. Perfect, in my opinion, for a good brief-communication on the event. At page 7, line 16, Authors refer to an "Amateur video footage"; it could be included in the supplementary materials, or, if it is published somewhere on internet, the related link could be reported.

The consideration about flow velocity (page 7, lines 19-22) are quite simple and not precise, as also stated by the Authors. I suggest removing it.

Section 7 is very long, with a lot of historical information; they are interesting, but not so useful to the discussion in a scientific paper. I suggest a huge revision and a huge reduction of this section. As an example, lines from 4 to 33 at page 8 could be reduced or removed.

Sections 8.1, 8.2, and 8.3 are interesting but very long. They could be reduced. I suggest to join sections 7 and 8 in a new section, namely a "Discussion" section, reducing them and keeping only the necessary parts. Finally, I suggest removing section 8.6.

Concerning the relation among typhoons, climate change and ground phenomena in tropical areas I suggest considering the work made by Chiang and Chang (2011). They
stated that in the western North Pacific, typhoons and rainfall rates are predicted to intensify because of climate change. As a result, they predict and increase in shallow landslides.

TECHNICAL CORRECTIONS

Everywhere in the text, I suggest using abbreviations for measure units, e.g., "km" instead of "kilometers", "m" instead of "meters", etc.

Page 3, Line 26: Check references. Some commas are missing and there are two closing parenthesis.

Figures: I would suggest using colours for the photos.

REFERENCES

Chang, S.-H., Chiang, K.-T., 2011. The potential impact of climate change on typhoon-triggered landslides in Taiwan, 2010–2099. Geomorphology 133, 143–151, doi:10.1016/j.geomorph.2010.12.028.

Cruden, D.M., Varnes, D.J., 1996. Landslide types and processes. In: Turner, A.K., Schuster, R.L. (Eds.) Landslides investigation and mitigation. Transportation research board, US National Research Council. Special Report 247, Washington, DC, Chapter 3, pp. 36–75.

Hungr, O., Leroueil, S., Picarelli, L., 2013. The Varnes classification of landslide types, an update. Landslides 11(2), 167-194, doi: 10.1007/s10346-013-0436-y.

Varnes, D.J., 1978. Slope movement types and processes. In: Schuster, R.L., Krizek, R.J. (Eds.) Landslides, analysis and control, special report 176: Transportation research board, National Academy of Sciences, Washington, DC., pp. 11–33.

---

## Short Comment (SC1) · 4 Jun 2016

We are grateful for the review by Referee #1, and accept many of its comments and suggestions. However, we believe Referee #1 interprets the NHESS definition of a research paper, and of "substantial and original scientific results" too narrowly. The judgment that our report "is not a scientific research paper" dismisses the effort expended in gathering remotely-sensed data to use for evaluating debris flow in the field, the fieldwork itself, and the scholarship that went into examining the literature in search for clues regarding what Typhoon Bopha and the Andap tragedy might signify for the

future.

As its name implies, the journal Natural Hazards and Earth System Sciences serves a very diverse audience of professionals concerned with natural hazards, and is the ideal home for this holistic examination of the Mayo debris flow, a major disaster triggered by the world's worst storm of 2012. Such catastrophes inextricably involve both natural and human components of the Earth system, and we evaluate it holistically as such. The NHESS website states: "Papers submitted to NHESS can address different techniques and approaches including theory, modelling, experiments, case studies, and instrumentations... Contributions dealing with multidisciplinary aspects of natural hazards and their consequences are welcome."

We document the world's seventh largest debris flow of record with all the meteorological, geological and historical reasons that made the Andap tragedy possible. (Typhoon Bopha deserves greater attention in its own right, because no Super Typhoon had ever formed and made landfall so close to the Equator.) We discovered that the literature concerned with future tropical-cyclone activity may not apply to low-latitude areas like Mindanao. Finding a major gap in our understanding is important too, for otherwise how can it be remediated?

One of our themes is how interacting geologic processes affect the development of communities. We illustrate how the lack of knowledge about fundamental principles of geomorphology, geologic history, and mass wasting can profoundly contribute to disasters. The NHESS website includes within the scope of the journal "...the analysis of the impact of climatic and environmental changes on natural hazards and their consequences", which is exactly what the last part of our paper tries to do.

Uncomfortable with our holistic approach, Reviewer #1 proposes to eliminate or greatly reducing several important parts of the manuscript:

A. Lines 5-25 on page 4, which describes the debris flow phenomenon. We believe it is necessary because one cause of the tragedy was precisely that many land use plan-
ners, not only those who established New Bataan and Andap, are still unacquainted with debris flows. Are not such decision makers a target audience for NHESS? Other readers not well grounded in geology might also find this useful.

B. Lines 19-22 on page 7, which explain how we estimated the velocity of the debris flow that hit Andap. Omitting this, however, would have us simply assert to the uninformed the velocity, without giving our basis for arriving at it.

C. Section 5 [Geomorphologic setting and history of New Bataan and the Mayo debris flow], It is rare that the entire history of a community from its establishment to its devastation is available, and that events and decisions made along the way contributed to the disaster can be described, as these 27 lines do.

D. The review characterizes Section 7, The role of Philippine population growth, as "interesting, but not so useful to the discussion in a scientific paper". That unabated growth motivated the establishment of New Bataan and Andap, and continues to exacerbate the hazards in a country where safe areas to develop are already virtually nonexistent.

E. Virtually our entire Section 8, our exhaustive evaluation of what the climatologic literature might tell us about future typhoon impacts on Mindanao. We cannot simply substitute it for the assertion by Chang and Chiang (2011) that typhoon activity will increase in Taiwan. See for example the comprehensive review bv Knutson and his nine coauthors (2010), which is much more equivocal. This is especially true for Mindanao and other low-latitude areas, which, even though they have large populations, are given short shrift by most meteorological and climatologic analyses.

F. Finally, Reviewer #1 suggests that we excise Section 8.6, Other climate-related hazards in the Philippines and Mindanao. We think these six lines are necessary to inform the readership that Mindanao is susceptible to other hazards related to climate change.

First specific comment: We will modify lines 25-27 on page 1 to state that published

definitions also refer to debris flows as a variety of landslide, and will cite the sources provided by the reviewer.

Second specific comment: We will prepare a figure that shows the high-resolution imagery that we used to map out the debris flow deposit.

A very well taken suggestion is that we provide internet links for the amateur video footage of the debris flow we mention on page 7, line 16, which we will do.

Reviewer #1's TECHNICAL CORRECTIONS are also very welcome, and the manuscript will be modified accordingly, with many thanks.
* * *

---

## Short Comment (SC2) · 8 Jun 2016

I might add to Dr. Rodolfo's reply to Referee #1 that our paper, as written, is entirely harmonious with NHESS's own home page description:

"Natural Hazards and Earth System Sciences (NHESS) is an interdisciplinary and international journal dedicated to the public discussion and open-access publication of high-quality studies and original research on natural hazards and their consequences. Embracing a holistic Earth system science approach, NHESS serves a wide and diverse community of research scientists, practitioners, and decision makers concerned

with detection of natural hazards, monitoring and modelling, vulnerability and risk assessment, and the design and implementation of mitigation and adaptation strategies, including economical, societal, and educational aspects."

Integrating earth science with the socio-political aspect of disaster research is sorely needed, more so in the Philippines where original scholarly work with this theme is very limited.

---

## Referee Comment (RC2) · Anonymous Referee #2 · 11 Jun 2016

General comments The paper deals with a hot topic and gives a thorough insight into a major rainfall-induced instability event occurred in the Philippines. Moreover, it is very well written and its structure is consistently logical and coherent.

Specific comments 1) Considering the importance that correctly authors confer to the classification of the event occurred at Mindanao in 2012, such issue deserves a major attention. Authors, in fact, reject the inclusion of the observed phenomena among the categories of flash floods, hyperconcentrated flows or landslides. However, taking into account classification schemes different from those considered by authors, and

equally consolidated among gemorphologists, engineering-geologists and hydraulic engineers, the features described and commented could also be referred to (at least, partly) to gravitational slope instability mechanisms. To this respect, authors are invited to reconsider their position, referring to the following papers of general interest:

Cruden D.M., Varnes D.J. (1996) - Landslide types and processes. In: Turner A.K., Schuster R.L. (eds), Landslides. Investigation and mitigation. Transp. Res. Board, Nat. Res. Council, spec. Rep. 247, National Academy Press, Washington, D.C., 36-75.

Hungr O., Evans S.G., Bovis M., Hutchinson J.N. (2001) - Review of the classification of landslides of the flow type. Environmental and Engineering Geoscience, 7, 221-238.

Pierson T.C., Costa J.E. (1987) - A rheologic classification of subaerial sediment-water flows. In: Costa J.E, Wieczorek G.F. (eds) Debris flows/avalanches: process, recognition, and mitigation. Geol. Soc. Am., Reviews in Eng. Geol., 7, 1-12.

2) A second point which needs to be treated in a wider way is the shared opinion that landslides, and more generally geological risks have increased in the last decades also due to urbanization of unsafe areas. Unfortunately, this evidence cannot any longer be restricted to under-developed or developing countries. Several recent studies have demonstrated that also industrialized western nations suffer from similar phenomena, made possible by several reasons, (e.g. illegal housing actions). Authors are therefore invited to review their treatment of this topic: to this aim, useful hints can be found in the following papers.

Cascini L., Bonnard C., Corominas J., Jibson R., Montero-Olarte J. (2005) - Landslide hazard and risk zoning for urban planning and development. In: Hungr O. et al. (eds), Landslide Risk Management, 199-235, CRC Press.

Di Martire D., De Rosa M., Pesce V., Santangelo M.A., Calcaterra D. (2012) - Landslide hazard and land management in high-density urban areas of Campania region, Italy.

Nat. Hazards Earth Syst. Sci., 12, 905–926.

Lari S., Frattini P., Crosta G.B. (2012) - Local scale multiple risk assessment and uncertainty evaluation in a densely urbanised area (Brescia, Italy). Nat. Hazards Earth Syst. Sci., 12, 3387-3406.

Technical corrections

No technical corrections have been made

---

## Short Comment (SC3) · 13 Jun 2016

We thank Referee #2 for very useful comments. The manuscript is being amended to incorporate the comment and references about the definitions of landslides and debris flows, which was also mentioned by Referee #1.

We are also incorporating the second point that developing countries are also experiencing more frequent hazards owing to urbanization of unsafe areas, along with its reerences.

---

## Author Comment (AC1) · 1 Jul 2016

These are our final author comments on behalf of all co-authors (final response phase), all of which were already published on the NHESS page [http://www.nat-hazards-earth-syst-sci-discuss.net/nhess-2016-102/#discussion]:

1. Responses to Referee #1

i. From K S Rodolfo, published on 04 June 2016: We are grateful for the review by Referee #1, and accept many of its comments and suggestions. However, we believe Referee #1 interprets the NHESS definition of a research paper, and of "substantial and

original scientific results" too narrowly. The judgment that our report "is not a scientific research paper" dismisses the effort expended in gathering remotely-sensed data to use for evaluating debris flow in the field, the fieldwork itself, and the scholarship that went into examining the literature in search for clues regarding what Typhoon Bopha and the Andap tragedy might signify for the future.

As its name implies, the journal Natural Hazards and Earth System Sciences serves a very diverse audience of professionals concerned with natural hazards, and is the ideal home for this holistic examination of the Mayo debris flow, a major disaster triggered by the world's worst storm of 2012. Such catastrophes inextricably involve both natural and human components of the Earth system, and we evaluate it holistically as such. The NHESS website states: "Papers submitted to NHESS can address different techniques and approaches including theory, modelling, experiments, case studies, and instrumentations... Contributions dealing with multidisciplinary aspects of natural hazards and their consequences are welcome."

We document the world's seventh largest debris flow of record with all the meteorological, geological and historical reasons that made the Andap tragedy possible. (Typhoon Bopha deserves greater attention in its own right, because no Super Typhoon had ever formed and made landfall so close to the Equator.) We discovered that the literature concerned with future tropical-cyclone activity may not apply to low-latitude areas like Mindanao. Finding a major gap in our understanding is important too, for otherwise how can it be remediated?

One of our themes is how interacting geologic processes affect the development of communities. We illustrate how the lack of knowledge about fundamental principles of geomorphology, geologic history, and mass wasting can profoundly contribute to disasters. The NHESS website includes within the scope of the journal ": : :the analysis of the impact of climatic and environmental changes on natural hazards and their consequences", which is exactly what the last part of our paper tries to do. Uncomfortable with our holistic approach, Reviewer #1 proposes to eliminate or greatly reducing

several important parts of the manuscript:

A. Lines 5-25 on page 4, which describes the debris flow phenomenon. We believe it is necessary because one cause of the tragedy was precisely that many land use planners, not only those who established New Bataan and Andap, are still unacquainted with debris flows. Are not such decision makers a target audience for NHESS? Other readers not well grounded in geology might also find this useful.

B. Lines 19-22 on page 7, which explain how we estimated the velocity of the debris flow that hit Andap. Omitting this, however, would have us simply assert to the uninformed the velocity, without giving our basis for arriving at it.

C. Section 5 [Geomorphologic setting and history of New Bataan and the Mayo debris flow], It is rare that the entire history of a community from its establishment to its devastation is available, and that events and decisions made along the way contributed to the disaster can be described, as these 27 lines do.

D. The review characterizes Section 7, The role of Philippine population growth, as "interesting, but not so useful to the discussion in a scientific paper". That unabated growth motivated the establishment of New Bataan and Andap, and continues to exacerbate the hazards in a country where safe areas to develop are already virtually nonexistent.

E. Virtually our entire Section 8, our exhaustive evaluation of what the climatologic literature might tell us about future typhoon impacts on Mindanao. We cannot simply substitute it for the assertion by Chang and Chiang (2011) that typhoon activity will increase in Taiwan. See for example the comprehensive review bv Knutson and his nine coauthors (2010), which is much more equivocal. This is especially true for Mindanao and other low-latitude areas, which, even though they have large populations, are given short shrift by most meteorological and climatologic analyses.

F. Finally, Reviewer #1 suggests that we excise Section 8.6, Other climate-related hazards in the Philippines and Mindanao. We think these six lines are necessary to inform the readership that Mindanao is susceptible to other hazards related to climate change. First specific comment: We will modify lines 25-27 on page 1 to state that published definitions also refer to debris flows as a variety of landslide, and will cite the sources provided by the reviewer.

Second specific comment: We will prepare a figure that shows the high-resolution imagery that we used to map out the debris flow deposit.

A very well taken suggestion is that we provide internet links for the amateur video footage of the debris flow we mention on page 7, line 16, which we will do.

Reviewer #1's TECHNICAL CORRECTIONS are also very welcome, and the manuscript will be modified accordingly, with many thanks.

ii. From R. N. Eco, published 08 June 2016: I think the reviewer's definition of new scientific results as only new results or methods is too constricting. The theme of the paper is the interaction between geologic processes and how it affects the development of communities/society. Our research showed that knowledge, or lack thereof, of fundamental geologic principles such as geomorphology, geologic history, and mass wasting have profound effects on the occurrence of disasters. The NHESS website lists the following as one of the scope of the journal: "- the analysis of the impact of climatic and environmental changes on natural hazards and their consequences" I think our paper does this exactly.

Regarding the specific comments: Second comment: We will prepare a figure that shows high resolution imagery that we used to map out the debris flow deposit.

On the technical corrections: We'll make the necessary corrections to the manuscript.

Response to Referee #2 from K. S. Rodolfo, published 13 June 2016: We thank Referee #2 for very useful comments. The manuscript is being amended to incorporate the comment and references about the definitions of landslides and debris flows, which

was also mentioned by Referee #1. We are also incorporating the second point that developing countries are also experiencing more frequent hazards owing to urbanization of unsafe areas, along with its references.
* * *